# Ultrasound-Triggered Release of 5-Fluorouracil from Soy Lecithin Echogenic Liposomes

**DOI:** 10.3390/pharmaceutics13060821

**Published:** 2021-06-01

**Authors:** Charles Izuchukwu Ezekiel, Alain Murhimalika Bapolisi, Roderick Bryan Walker, Rui Werner Maçedo Krause

**Affiliations:** 1Department of Chemistry, Faculty of Science, Rhodes University, Makhanda 6140, Eastern Cape, South Africa; izuchukwuezekiel@yahoo.com (C.I.E.); alainbapo@gmail.com (A.M.B.); 2Division of Pharmaceutics, Faculty of Pharmacy, Rhodes University, Makhanda 6140, Eastern Cape, South Africa; r.b.walker@ru.ac.za; 3Center for Chemico and Biomedicinal Research, Rhodes University, Makhanda 6140, Eastern Cape, South Africa

**Keywords:** colorectal cancer, 5-fluorouracil, soy lecithin, liposomes, ultrasound, stimuli response

## Abstract

Colorectal cancer is the third most diagnosed cancer and the second leading cause of death. The use of 5-fluorouracil (5-FU) has been the major chemotherapeutic treatment for colorectal cancer patients. However, the efficacy of 5-FU is limited by drug resistance, and bone marrow toxicity through high-level expression of thymidylate synthase, justifying the need for improvement of the therapeutic index. In this study, the effects of ultrasound on echogenic 5-FU encapsulated crude soy liposomes were investigated for their potential to address these challenges. Liposomes were prepared by thin-film hydration using crude soy lecithin and cholesterol. Argon gas was entrapped in the liposomes for sonosensitivity (that is, responsiveness to ultrasound). The nanoparticles were characterized for particle size and morphology. The physicochemical properties were also evaluated using differential scanning calorimetry, Fourier transform infrared and X-ray diffraction. The release profile of 5-FU was assessed with and without 20 kHz low-frequency ultrasound waves at various amplitudes and exposure times. The result reveal that 5-FU-loaded liposomes were spherical with an encapsulation efficiency of approximately 60%. Approximately 65% of 5-FU was released at the highest amplitude and exposure time was investigated. The results are encouraging for the stimulated and controlled release of 5-FU for the management of colorectal cancer.

## 1. Introduction

Colorectal cancer (CRC), also known as bowel cancer, is a type of cancer affecting the colon and the rectum. It is the third most commonly diagnosed cancer in men and second in women, and is the second leading cause of death worldwide. Indications are that the incidence of colorectal cancer is increasing [1]. 5-Fluorouracil (5-FU), either used alone or in combination with other chemotherapy regimens, has been the major therapeutic agent for the treatment of CRC in patients, in addition to patients with some breast cancer or head and neck cancer [2]. Research has shown that 5-FU exerts an anti-cancer effect through thymidylate synthase (TS) inhibition, the pathways for which have not been completely explained. High-level expression of TS results in 5-FU resistance [3]. Although this drug has been in use for more than 40 years and remains an important chemotherapeutic agent, it exhibits serious side effects, including gastrointestinal and bone marrow toxicity [4]. In addition, it is associated with the development of resistance, resulting in its lack of efficacy [5]. 5-FU is difficult to deliver as a result of its hydrophilicity or low lipophilicity. This gives rise to low bioavailability when administered rectally or orally. These delivery challenges and reduction of toxic side effects have been addressed, in part, by encapsulating 5-FU into vesicular nanocarriers such as liposomes which effectively controlled the pharmacokinetic properties of the drug, delivering the medication within the desired therapeutic range to tumor cells without affecting healthy cells [6,7].

Liposomes are colloidal phospholipid vesicles, usually in the nano or micron range, made up of one or several lipid layers enclosing an aqueous core. They have been extensively studied and applied in drug delivery applications due, partly, to favorable properties such as low toxicity, good biocompatibility, biodegradability and the ability to encapsulate hydrophilic and hydrophobic substances in the aqueous core and lipid bilayers, respectively [8]. These lipid components have been shown to encapsulate some gases, making them responsive to ultrasound stimulation. Indeed, some lipid-oriented ultrasound contrast agents are commercially available using this approach [9]. A type of nanocarrier, stealth liposomes have been developed that respond and release their payload when triggered with alternating magnetic fields [10]. Trucillo and co-workers have recently published a detailed review on the advances of liposomes from conventional methods of production and application to the supercritical approach of triggering and targeting [11]. While it is often hard to study these systems, gases are typically assumed to be entrapped between the lipid layers after introduction in the manufacturing process, thus behaving similarly to hydrophobic drugs [12]. A liposomal formulation encapsulating both gases and therapeutic agents would therefore be responsive to ultrasound and exhibit ultrasound-triggered release. It has been demonstrated that gases such as nitric oxide (NO) can be encapsulated into echogenic liposomes, thereby producing a ‘theranostic’ agent with both echogenic and bioactive gas-delivery characteristics [13].

Ultrasound use in the clinic setting is well known and its use includes imaging, tumor and fibroid shattering, kidney stone ablation and physiotherapy treatments, amongst others. The therapeutic use of ultrasound is usually due to either mechanical or heat effects [14]. The application of ultrasound (US) to echogenic liposomes may increase the delivery of chemotherapeutic drugs to tumor cells. In addition, the simultaneous use of ultrasound for therapy and diagnosis is in line with the emerging trend of ‘theranostics,’ for which the benefits include fewer multi-step procedures and improved patient care [15]. The mechanism of activity lies in the potential of the US to enhance the permeabilization of liposome bilayers through the creation and oscillation of gas bubbles in a liquid medium by either sustained growth and oscillations over many acoustic cycles (stable cavitation) or violent growth and collapse in less than a cycle (transient or inertial cavitation) [16]. The disruption of biological membranes and liposomes by ultrasound could occur by either stable or inertial cavitation [17,18]. Cavitation induces the release of the encapsulated therapeutic at the cancer site, increasing tumor cell uptake of the drug [19]. However, total reliance on membrane permeabilization on cavitation or ultrasound is not yet fully understood. Ultrasound has been explored to trigger the release of therapeutics from various carriers [12,20,21] such as lipospheres, polymer systems or liposomes made from synthetic phospholipids. To the best of our knowledge, the use of crude soy lecithin (CSL) in an echogenic cancer study has not yet been reported. CSL is a naturally occurring lipid mixture derived from soybeans, which is relatively cheap in comparison to synthetic phospholipids and has been shown to be advantageous for the production of liposomes for the delivery of other therapeutic agents [22].

Previous work in our laboratory has focused on methods for formulating and stabilizing liposomes using crude soy lecithin for encapsulating a wide range of therapeutic drugs, including isoniazid, rifampicin, efavirenz, proteins and photosensitizers. Our studies have also evaluated the physicochemical properties and in vitro release profiles of drugs in the carrier technology and included evaluation of the susceptibility to stimulated release using pH responsiveness [22,23,24,25,26]. However, the effect of physical stimuli such as ultrasound on the release of anti-cancer drugs from these promising cost-effective nanocarriers has not yet been explored.

This report details the formulation, preparation and physicochemical characterization of crude soybean echogenic liposomes encapsulating 5-FU. The carriers were produced using the thin-film hydration method and pressure freeze technology to form vesicles and passively entrap the drug and an inert gas. This in vitro study investigated the effects of ultrasound parameters on the release of 5-FU drug from the novel carrier. Cell disruption by cavitation has been shown to be inversely related to the frequency of ultrasound [27]. Low-frequency ultrasound should be more successful in increasing membrane permeability [16]. As a result of this, we designed this study to focus on the use of 20 kHz low-frequency ultrasound to evaluate the release of 5-FU with and without the application of ultrasound. The data derived from this study may aid the optimization of ultrasound parameters for in vivo triggered drug release to colorectal cancer tumors and sites.

## 2. Materials and Methods

### 2.1. Materials

The crude soybean lecithin granules used in this work were purchased from Health Connection Wholefoods (USA) and were used as procured without any purification. From the manufacturer’s information, 100 g of this granule comprises 23 g of phosphatidylcholines, 14 g of phosphatidylinositol, 35 g of unsaturated fats, 13 g of saturated fats, 8 g of glycemic carbohydrates and 0.11 g of sodium. Cholesterol was purchased from Carlo Erba/Divisione Chimica (Milan, Italy). 5-Fluorouracil, d-mannitol, mono- and dibasic sodium phosphate, methanol, acetonitrile (HPLC grade) and chloroform were purchased from Sigma-Aldrich (Steinheim, Germany). Argon gas from Afrox (Johannesburg, South Africa) of 99.9% purity was used. HPLC-grade 18 Ωohm ultra-pure water prepared using a Milli-Q academic A10 water purification system (Millipore^®^, Bedford, MA, USA) was used throughout the study. A Heidolph Hei-VAP rotary evaporator was used at 30 mm Hg pressure and 60 °C water bath temperature. A LABCONCO FreeZone^®^ 6 Liter Benchtop Freeze Dry System (Kansas City, MO, USA) was used in freeze drying the formulations. An Agilent 1100 liquid chromatography system equipped with a quaternary pump, degasser, diode array detector (DAD) and manual injector with a Phenomenex^®^ Luna LC Column (5 µ C18, 100 Å, 250 × 4.6 mm i.d.) was used for analysis. A DSC-6000 was used for thermal analysis. A PerkinElmer Spectrum 100-FT-IR spectrometer was used to record IR spectra while an XRD D8 Discover (Bruker, Billerica, MA, USA) was used to characterize the crystallinity of the liposomal formulations. The particle size, polydispersity index and zeta potential were monitored using Zetasizer nano ZEN-3000 MAL1043132 from Malvern Instruments (Malvern, UK). For the morphology study, a Zeiss Libra-120KV TEM (Oberkochen, Germany) was used with samples deposited on a holey-carbon copper. The ultrasound system used for the triggering release was a Sonopuls HD 4200 high-power ultrasound from Bandelin (Berlin, Germany).

### 2.2. Methods

#### 2.2.1. Preparation of 5-FU Encapsulated Liposomes and Determination of the Encapsulation Efficiency

The carrier formulations were prepared using a conventional thin-film hydration method in which 100 mg of the lipid components comprising crude soybean lecithin (75 mg) and cholesterol (25 mg) were dissolved in 1 mL chloroform in a clean 25 mL round bottom flask and dried using a rotary evaporator set at 60 °C under a pressure of 30 mm Hg at 250 revolutions per minute (rpm) for 5 min. The thin film was removed and stored overnight in a desiccator at room temperature of 22 ± 3 °C for complete drying. A sample (50 mg) of 5-FU was dissolved in 5 mL of pH 7.4 phosphate buffer and added to the thin films, and allowed to passively load with hydration over 60 min under continuous stirring at 400 rpm and at 60 °C using a magnetic stirrer and a hot plate. The suspension was further homogenized for 30 min using an ultrasonic bath sonicator and then vortexed for 5 min. The liposomes produced were allowed to equilibrate at room temperature of 25 °C, and HPLC grade water was added to make up the volume to 20 mL for the purpose of centrifuging. Empty or blank liposomes and free 5-FU were also treated in a similar manner to serve as positive and negative controls, respectively.

The 5-FU-loaded liposomal formulation was transferred to a 50 mL centrifuge tube and centrifuged for 20 min using an MSE Mistral-1000 low-speed centrifuge set at a relative centrifugal force (RCF) of 1020 g. The pellet (unencapsulated 5-FU) was separated, and the supernatant was transferred into 1.5 mL Eppendorf tubes and centrifuged using an Eppendorf 5414 micro high-speed centrifuge set at an RCF of 15,600× *g* for 20 min. The supernatant was discarded and the liposomes in the pellet rinsed with 1 mL HPLC grade water three times to remove any unencapsulated 5-FU. The 5-FU-loaded liposomes were then freeze dried for 48 h using the Labconco freeze dryer. The freeze-drying process was conducted by freezing the samples in a vial at −196 °C using liquid nitrogen for 15 min, after which primary drying at −50 °C under a vacuum of 0.315 Torr for 36 h was implemented. Secondary drying at 25 °C for 12 h was then performed. The vials were immediately removed after freezing and sealed with rubber caps and were stored at 4 °C prior to further characterization. A validated HPLC method was used to quantify the amount of 5-FU in the supernatant (Appendix A). The solutions were filtered with a 0.22 µm PVDF Millipore syringe filter prior to analysis. The amount of 5-FU incorporated into the nanocarrier was determined quantitatively using the validated HPLC method. The quantitative analysis was performed in triplicate and, for each replicate, three injections of 20 µL were done into the HPLC instrument according to the method described in the Appendix A. The experiments were replicated (*n* = 3). The encapsulation efficiency (%EE) was determined using the following equation [28].
(1)$$\%\mathrm{EE}=\frac{\mathrm{Mass}\;\mathrm{ofEncapsulated}\;5-\mathrm{FU}}{\mathrm{Total}\;\mathrm{mass}\;\mathrm{of}\;5-\mathrm{FU}\;\mathrm{used}}\;\times \;100$$

#### 2.2.2. Preparation of Echogenic Liposomes

The same amounts of the lipid components as mentioned in Section 2.2.1 were weighed and dissolved in 1 mL of chloroform and dried, using the method described by Huang et al. with some modification of the lipid composition and drug concentration [29]. Briefly, 50 mg 5-FU were dissolved in a solution of 0.36 mol/L d-mannitol as it facilitates physiological osmolarity and entrapment of Ar and 5-FU. The suspension was hydrated and sonicated as described in Section 2.2.1 and then transferred into a 50 mL high-pressure resistance glass vial capped with a suba-seal rubber septum. The Ar was introduced into the vial through the septum with a 10 mL gas syringe with a 27G × 1/2″ needle. The septa were tested for leakage for 6 h, and no leak was detected. The high-pressured liposomal formulation was incubated at room temperature for 30 min and then frozen by cooling to −196 °C in liquid nitrogen for 15 min. As soon as the formulation was removed from the cooling system, the pressure was released by removing the septum, and the depressurized frozen formulation subsequently thawed at room temperature of 22 ± 3 °C for 20 min. These experiments were replicated (*n* = 3). The encapsulation efficiency of liposomes for 5-FU was determined according to the procedure described in Section 2.2.1.

#### 2.2.3. Particle Size and Zeta Potential Characterization

The particle size and zeta potential of the formulations were measured by dynamic light scattering. A sample (2 mg) of the freeze-dried powdered formulation was re-dispersed in 1 mL HPLC grade water. The suspension was gently shaken and allowed to equilibrate at room temperature of 25 °C for 3 h and the measurements were performed in triplicate for each of the samples at a scattering angle of 173° and 25 °C. For each measurement, 10 runs (of 10 s each) were performed. The particle size distribution was confirmed by analyzing TEM images with ImageJ software.

#### 2.2.4. Liposome Morphology

The shapes of the liposomes were confirmed using transmission electron microscopy (TEM). The same aqueous sample used for particle size and zeta potential determination was first spotted on a copper grid and allowed to air dry for 10–15 min on a filter paper, the uranyl acetate was then spotted on the grid containing the samples and allowed to dry out at room temperature for 24 h before the TEM analysis.

#### 2.2.5. Differential Scanning Calorimetry (DSC)

The thermal behavior of the free 5-FU was evaluated and compared with the liposomal 5-FU. The samples were heated from 20–350 °C at a heating rate of 10 °C/min in a nitrogen-saturated environment set at 20 mL/min. An empty aluminum pan was used as a reference, and the thermogram showing upward endothermic peaks, together with the respective heat flow curves, was recorded and analyzed using the Pyris DSC software series.

#### 2.2.6. X-ray Diffraction (XRD)

The crystalline nature of the 5-FU liposomes, blank liposomes and free 5-FU was assessed using XRD. The evaluation was done with a nickel filter and Cu-Kα radiation set at 1.5404 Å while running the scan at 2–ϴ range 10° to 60°, the scanning speed was 1°/min and the slit width was 6.0 mm.

#### 2.2.7. Fourier Transform Infrared Spectroscopy (FTIR)

The IR spectra of the liposomal 5-FU, empty liposomes and free 5-FU were recorded in an attenuated total reflection mode. Eight scans were performed for each of the samples in a wavenumber range of 4000 cm^−1^ to 500 cm^−1^.

#### 2.2.8. In Vitro Release of 5-FU

The in vitro release of 5-FU was compared to that from liposomes containing 5-FU in HPLC grade water. The 5-FU content was determined using the method described by Nkanga et al. [30], with some changes in the aliquot and concentrations considered.

A sample of freeze-dried liposomal 5-FU (20 mg) or free 5-FU (2 mg) was dissolved in 2 mL HPLC grade water and allowed to incubate for 60 min while the suspension was occasionally gently shaken by hand for homogenization. An aliquot of the suspension (0.5 mL) was placed in a 5 mL volumetric flask, and acetonitrile was added to make up the volume. The mixture was sonicated for 30 min at 60 °C to destroy the liposome structure and filtered with a 0.45 µm Millipore filter, and the solution was analyzed using the validated HPLC method to determine the total available concentration of 5-FU.

Another 0.5 mL of the dissolved liposomal suspension and free 5-FU were transferred into the dialysis tubing membrane (Membra-Cell MD10 14 × 100 CLR, Sigma-Aldrich, St. Louis, MI, USA) for the release studies. The dialysis bags were sealed and placed in a glass vial containing 20 mL of pH 7.4 phosphate buffer and maintained at 37 °C under continuous stirring at 100 rpm all through the duration of the study. An aliquot of 5 mL of the sample was withdrawn after 0.5, 1, 1.5, 2, 3, 4, 5, 9 and 12 h for the quantification of the 5-FU according to the HPLC validated method (Appendix A). For the renewal of the release medium, an equivalent volume of the fresh buffer was added after every sampling. These experiments were replicated (*n* = 3).

#### 2.2.9. Ultrasound-Triggered Release

The effect of ultrasound stimulation on the release profile of the drug was evaluated using the method described in Section 2.2.10 with the addition of further processes. The dialysis bag containing the dissolved echogenic liposomes was inserted in a 50 mL glass beaker containing 30 mL of PBS buffer pH 7.4. A 20 kHz low-frequency ultrasonic processor (Sonopuls HD 4200, Bandelin, Berlin, Germany) was used. The 13 mm ultrasonic probe was immersed in the glass beaker containing the sample. The beaker was placed in a temperature-regulated water bath and the temperature was monitored at 37 °C all through the duration of the experiment to avoid hyperthermia-induced drug release [31]. The triggering was conducted in a continuous mode at varying amplitudes and exposure times [32]. The sample was immediately removed from the release medium after the ultrasound irradiation, and the released drug was evaluated from the medium. A fresh sample was prepared according to the above procedure to evaluate the effects of ultrasound exposure on the drug percentage of cumulative release. The sample in the dialysis bag was placed in a glass vial containing 30 mL of the release medium and exposed to ultrasound irradiation of various parameters. After the ultrasound stimulation, the sample was maintained at 37 °C and 100 rpm and samples were withdrawn according to the time intervals in Section 2.2.10. With respect to the ultrasound parameters selected, a total of 7 groups were investigated, including the control group, which was non-echogenic liposomes, and was not exposed to ultrasound irradiation as with the other 6 groups. The remaining liposomes were triggered for 5 min or 10 min at amplitudes of 10%, 15% and 20%. These experiments were replicated (*n* = 3). The 5-FU released was quantified using the validated HPLC method (Appendix A).

#### 2.2.10. Stability Studies

The freeze-dried echogenic liposomal formulations were stored at 4 °C, and their stability was monitored for four weeks. Two milligrams of the samples were dissolved in 1 mL phosphate buffer pH 7.4, and aggregation and fusion were studied through the variations in zeta potential for day 7, day 14, day 21 and day 28 with the use of DLS. These experiments were done in triplicate.

#### 2.2.11. Statistical Analysis

We replicated all our experiments (*n* = 3) and report the data as mean ± standard deviation (SD). We used Minitab 17 (Minitab, Ltd., Coventry, UK) for our statistical analysis and applied one-way ANOVA for our comparative data analysis. We considered our data statistically significant when the *p*-value was ˂0.05. We used ImageJ software to process TEM images and OriginPro 9 to plot graphs.

## 3. Results and Discussion

### 3.1. Encapsulation Efficiency (EE)

The encapsulation efficiency of drug-loaded liposomes is an important characteristic for the effectiveness of the method used for encapsulation. In this study, the formulation with the lipid to drug mass ratio of 2:1 (100 mg lipid components and 50 mg of 5-FU) gave the highest encapsulation efficiency of 62 ± 2%, as presented in Table 1. These data agree with some reports in which the %EE of 5-FU-loaded nanoparticles is described. Yu-Ling Fan et al. reported an %EE of 55 ± 1% for 5-FU encapsulated lecithin nanoparticles [33]. Yassin et al. also reported an %EE of 59% for 5-FU-loaded solid lipid nanoparticles [34]. However, Lopes et al. recorded 31% %EE of 5-FU in liposomes made of synthetic phospholipids and attributed it to the concentration of 5-FU used in the entrapment process [35]. This high %EE may also be attributed to the presence of carbohydrates, hydrophilic compounds in the soy lecithin liposomes, which can stabilize the 5-FU. As a hydrophilic drug, 5-FU is assumed to be entrapped in the aqueous core of the liposomal formulation [22].

### 3.2. Particle Size and Zeta Potential

The size distribution of the particles from the TEM image obtained without staining (Figure 1B) showed a Gaussian distribution with a peak at approximately 120 nm (Figure 1C), correlating with the particle size distribution by number derived from the evaluation of the same sample by DLS (Figure 1A). All the formulations showed a negative zeta potential in the range of −54 to −63 mV (Table 1). Particle size and zeta potential are two important parameters for maintaining in vivo integrity and behavior of nanoparticles. These observed negative zeta potentials are in line with what has been reported in previous studies using crude soy lecithin and a thin-film hydration method [23,30].

### 3.3. Particle Shape

The TEM images obtained after staining the samples with uranyl acetate (Figure 2) showed the spherical shape of the 5-FU encapsulated liposomes with little or no fusion or aggregation observed. This could be explained by the high negative zeta potential observed. This result matches DLS measurements of the samples. The zeta potential controls the stability of the liposomal formulations under physiological conditions [36]. However, further real-time studies would be necessary to confirm the stability of these formulations under physiological conditions.

### 3.4. Differential Scanning Calorimetry (DSC)

The DSC thermogram of free 5-FU showed a sharp melting endothermic peak at about 287 °C, close to the melting point of the compound (283 °C). The melting peak at about 287 °C was followed by decomposition, as reported in the literature [4]. The apparent disappearance of the endothermic peak in the liposomal 5-FU confirms the encapsulation of 5-FU and suggests conversion to an amorphous form (Figure 3). The endothermic transitions between 100 and 170 °C observed for the empty liposomes and the liposomal 5-FU are likely a melting endotherm for the phospholipids [37].

### 3.5. X-ray Diffraction (XRD)

The X-ray diffraction patterns of free 5-FU, liposomal 5-FU and empty liposomes are shown in Figure 4. The XRD pattern of free 5-FU showed a sharp and intense peak at 29° and smaller peaks at 16° and 33°, confirming its crystalline nature as also reported in the literature [4]. However, the empty liposomes showed a broad, amorphous profile. The peak at 29° could also be seen in the diffractogram of liposomal 5-FU, although at a lower intensity, suggesting that the drug was successfully encapsulated in the carrier. These data support the earlier DSC results that there was a change in the physical structure of 5-FU in liposomes, probably in a more amorphous state due to the lower intensity of the peak.

### 3.6. Fourier Transform Infrared Spectroscopy

The interactions of the free 5-FU, liposomal 5-FU and empty liposomes were studied using FT-IR. As can be seen in Figure 5, characteristic FTIR absorption bands of 5-FU can be found in the liposomal 5-FU. The characteristic band at 1660 cm^−1^ in the spectrum of free 5-FU is the stretching vibration of the C=O group [38]. The bands at 1060 and 1235 cm^−1^ in the liposome formulations can be related to the stretching vibration of the P=O in the phosphate group. The bands observed at about 2750 cm^−1^ in all the formulations are the C-H stretching bands. The characteristic peak at 1660 cm^−1^ in the free 5-FU corresponding to the C=O stretching vibration shifted to about 1680 cm^−1^ in the liposomal 5-FU. The spectral results further show that the drug was successfully encapsulated in the nanoparticles.

### 3.7. In Vitro Release of 5-FU

The in vitro release profiles of 5-FU from liposomes and free 5-FU were used to elucidate the release behavior of the drug. The cumulative percentage of 5-FU release from the liposomes in the first 2 h was approximately 50%, whereas that of free 5-FU was 70% (Figure 6). Encapsulation of 5-FU in crude soy lecithin liposome reduced both the rate and cumulative amount of 5-FU released. The burst effect after the first 2 h and slower phase of release is evident. These findings are in agreement with what has been reported for the release of 5-FU from a liposomal dispersion [39]. The total amount of 5-FU released from the liposomal dispersion was approximately 70% after 12 h, whereas 95% of the free 5-FU was dissolved after the same time. The release of free 5-FU was faster than that in the liposomal 5-FU; therefore, crude soy lecithin liposomal 5-FU may be explored as an effective nanocarrier for controlled release of therapeutics into colorectal cancer sites.

### 3.8. Effects of Ultrasound Amplitude and Exposure Time on 5-FU Release

Drug release from liposomes is affected by applied ultrasound [40]. We investigated amplitudes of 10%, 15% and 20% and the 5-FU released increased in proportion to the increase in applied amplitude, which could be a result of the introduction of transient cavitation in the liquid release medium (Figure 7). This cavitation might have occurred near the liposomal membrane or through small cavitation nuclei in the liposome aqueous core. The liposomal formulations were irradiated for an exposure time of 5 min and 10 min. We inferred that the higher the ultrasound amplitude and the longer the exposure time, the greater the amount of energy density transferred to the nanocarrier, which directly altered the quantity of 5-FU released. These data show that a large proportion of 5-FU can be released from the liposomes on exposure to low-frequency ultrasound applied over a short time. This could result in the availability of a higher concentration of the compound in the cancer cells when triggered, thereby resulting in prolonged circulation time with reduced off-target toxicity.

Figure 8 also shows the cumulative release rate of the drug on exposure to ultrasound. The rate of release after 12 h was above 95% for all the ultrasound parameters considered but was about 70% for the control. This showed that the cumulative percentage of 5-FU release from the echogenic liposomes increased significantly when previously triggered with ultrasound. The formulation triggered with a 20% amplitude for 10 min reached about 99% release after 5 h, confirming further that the higher the amplitude and exposure time, the greater the percentage release. We hypothesize that this could be a result of higher energy that went into the liposome, permeabilizing it and creating more pores. We assumed that the drug was released from the pores formed as a result of transient cavitation that occurs on ultrasound irradiation, and there was an exponential burst release after 2 h, followed by a gradual release phase. All the echogenic liposomes displayed a several-fold increase in the extent of drug release compared to the control that served as a reference. The complete mechanism behind ultrasound-triggered drug release from liposomes has not been fully explained in the literature. This, to an extent, might be due to some limitations in the methods of determining reversible changes in the bilayers of phospholipids that only occur during ultrasound irradiation. Our results correlate with what was reported by Abed et al. when they applied 3 MHz ultrasound to magnetic polylactic co-glycolic acid nanoparticles encapsulating 5-FU; they reported that an increase in the percentage of drug released is directly proportional to the exposure time and intensity [41]. Schroeder et al. also obtained a similar result when they controlled the release of doxorubicin, methylprednisolone hemisuccinate and cisplatin co-loaded in liposome using 20 kHz low-frequency ultrasound; they obtained about 80% release within a 3 min exposure. They concluded that the percentage of drug release was a function of amplitude and exposure time and also suggested that exposure of liposomes to ultrasound induced reversible pore-like defects in the membrane which allowed drugs to be released [40]. As displayed in our ultrasound system, the amount of energy delivered into the medium on ultrasound irradiation by the piezoelectric effect was observed to be higher with more prolonged exposure. In our experiment, for 5 min of exposure time and 20% amplitude, 39.428 KJ of energy was delivered to the medium while 75.315 KJ of energy was delivered to the medium on 10 min irradiation, and 64% of the drug was released. It can be concluded that the longer the exposure time, the higher the amount of energy imparted to the medium, causing more perturbation in the nanoparticle membrane and accelerating its rate of drug release.

### 3.9. Effect of Gas Entrapment on the Sensitivity of Liposomes to Ultrasound

We went further to investigate the effect of gas entrapment on the percentage release of the drug from the nanocarriers. We found that for the echogenic liposome (Elip 1), 38 ± 4% of the drug was released on exposure to a 10% amplitude of ultrasound for 10 min, while 16 ± 2% was released from the non-echogenic liposome (FE 1) (without a gas). This is a significant increase showing the impact of gas entrapment on the carrier’s sensitivity to ultrasound triggering. The non-echogenic liposome could release such a small percentage because of mechanical effects imparted on the carrier by the ultrasound and the atmospheric air inadvertently entrapped in the bilayer during formulation [42].

### 3.10. Stability Studies of Echogenic Liposomes Stored at 4 °C

As shown in Figure 9, all the echogenic liposomes studied for stability maintained their zeta potential within the range of −47 to −64 mV from the first day to the 28th day. This confirmed that the nanocarriers were stable for the period under storage at 4 °C as no fusion or aggregation was observed. The addition of d-mannitol in the formulation, which serves as a cryoprotectant, and the presence of carbohydrates in the phospholipids, were assumed to contribute to the echogenic liposomal formulations’ stability.

The stable ZP further proves our carrier’s appropriateness for triggered and control release of the therapeutics. It correlates with the study where chitosan proved that its interaction with negatively charged drugs, when entrapped into films, affected the drug release behaviors and the physicochemical characteristics of the drug and the polymer [43]. The zeta potential of nanoparticles affects the stability, release rate, circulation in the bloodstream and absorption across biological membranes [44]. Although our results have revealed an encouraging stability profile at 4 °C, ongoing research in our laboratories continues with the aim of elucidating the behaviors of echogenic liposomes using in vitro studies under physiological conditions. In particular, we are working with an expert to determine the effect of the ultrasound on cancer cells. Some early work indicates that the mechanical effects may stimulate some immune response.

## 4. Conclusions

5-Fluorouracil and argon were successfully encapsulated in crude soy lecithin echogenic liposomes produced by thin-film hydration. The liposomes that formed were spherical and of nanosized dimensions. The amplitude and exposure time to ultrasound increased the percentage of 5-FU released. This novel and facile method of production of echogenic liposomes for ultrasound-triggered drug delivery suggests the potential of developing practical in vivo systems for the delivery of chemotherapeutics to treat colorectal cancer and requires further investigation.

The overall mechanism and performance of ultrasound-mediated disruption of liposome bilayers remains to be clarified, however, the reality that less energy is needed to achieve 5-FU release from echogenic liposomes compared to control liposomes may be of clinical importance since damage to healthy cells and tissues during treatment can be reduced. The release of this chemotherapeutic agent from liposomes following exposure to low-frequency ultrasound of selected amplitude and exposure time may help the conceptual development of the potential use of low-frequency ultrasound to trigger and control the release of drugs from liposomes. However, for clinical application, a focused ultrasound transducer with high frequency is likely to be required to enhance focusing ability [45].

This research, to the best of our knowledge, is the first to report the encapsulation of 5-fluorouracil in crude soy echogenic lecithin liposomes produced using thin-film hydration and our results reveal that liposomes produced from natural soy lecithin have the potential to be used for ultrasound-triggered drug delivery to treat colorectal cancer. Ongoing research in our laboratories includes investigating the effects of using other biocompatible gases such as perfluorocarbons on ultrasound sensitivity and synthesizing echogenic liposomes with theranostic capabilities. Moreover, in vitro/ex vivo studies assessing the physiological stability as well as the therapeutic behavior of the echogenic liposomes for colorectal cancer treatment are currently under consideration.

## Figures and Tables

**Figure 1 pharmaceutics-13-00821-f001:**
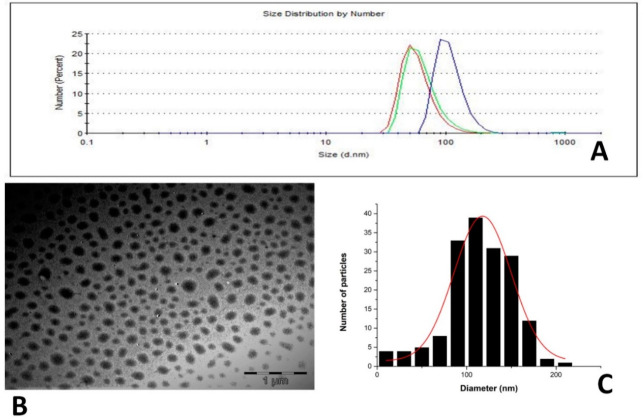
DLS size distribution by number (**A**), TEM image without staining (**B**) and size distribution histogram generated using ImageJ from the TEM image (**C**) of formulation FE 1.

**Figure 2 pharmaceutics-13-00821-f002:**
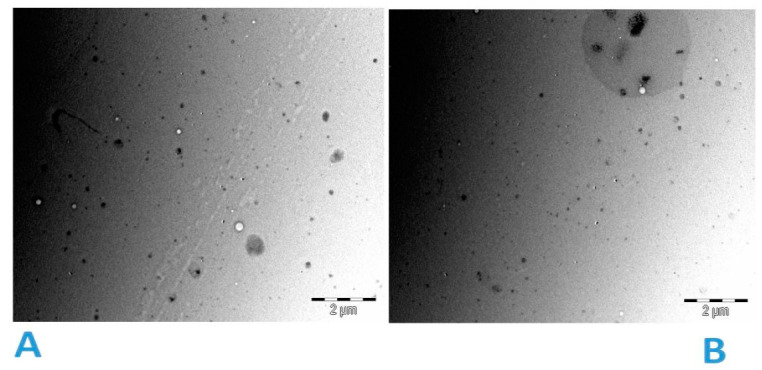
TEM images of liposomal 5-FU (**A**,**B**) after staining with uranyl acetate.

**Figure 3 pharmaceutics-13-00821-f003:**
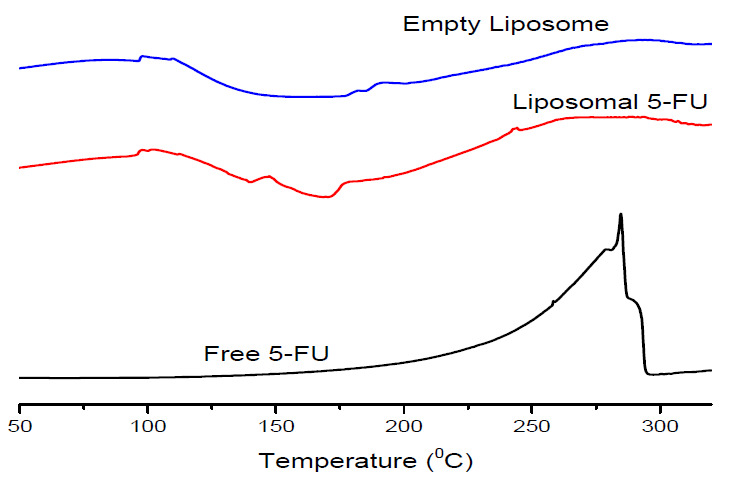
DSC thermogram of free 5-FU, liposomal 5-FU and blank liposomes.

**Figure 4 pharmaceutics-13-00821-f004:**
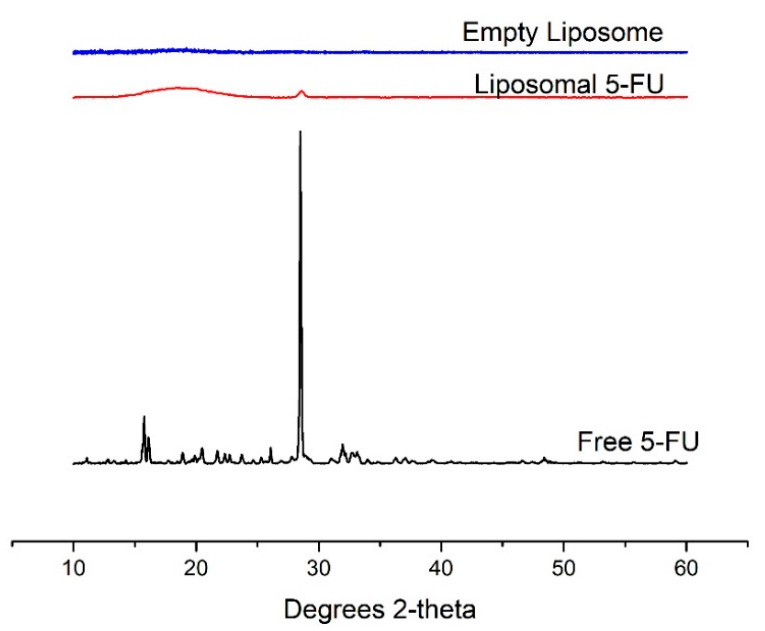
X-ray diffraction pattern of free 5-FU, liposomal 5-FU and empty liposomes.

**Figure 5 pharmaceutics-13-00821-f005:**
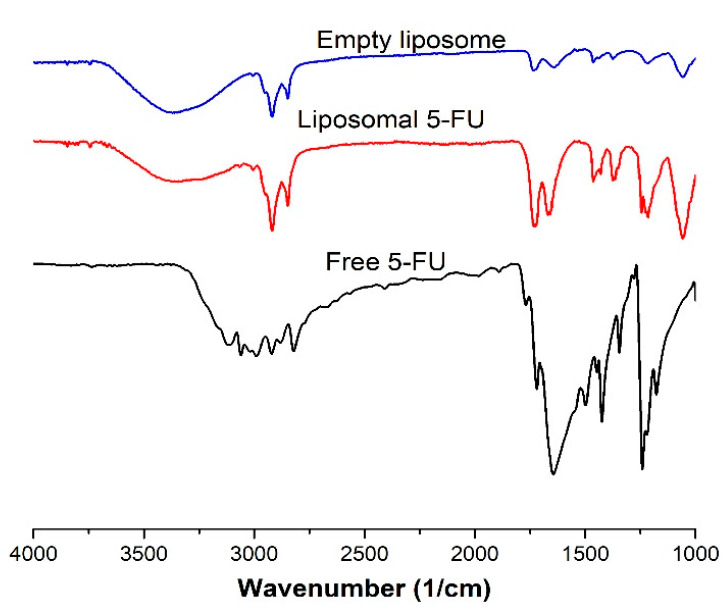
FT-IR spectra of free 5-FU, liposomal 5-FU and empty liposome.

**Figure 6 pharmaceutics-13-00821-f006:**
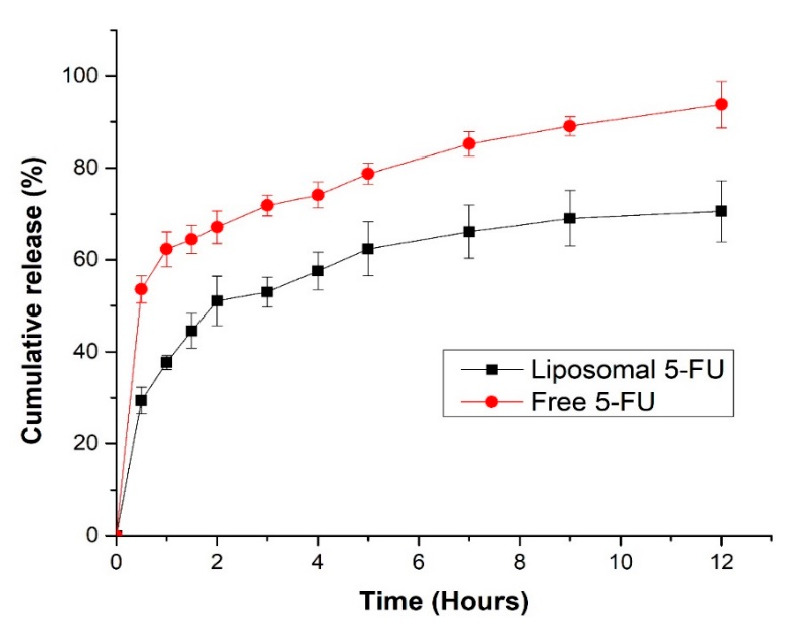
In vitro release profiles of liposomal 5-FU (squares) and free 5-FU (circles) without ultrasound application.

**Figure 7 pharmaceutics-13-00821-f007:**
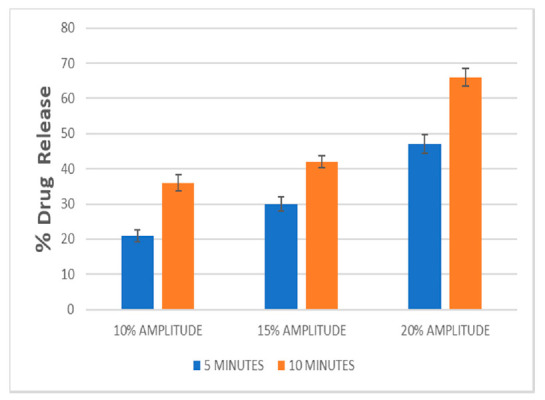
Release of 5-FU from echogenic liposomes exposed to 10 to 20% ultrasound amplitudes for 5 and 10 min.

**Figure 8 pharmaceutics-13-00821-f008:**
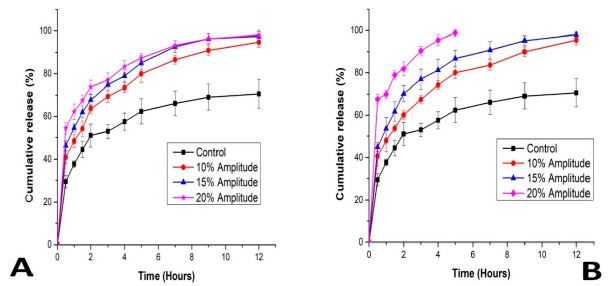
5-FU release from echogenic liposomes exposed to ultrasound of different amplitudes for 5 min (**A**), and 10 min (**B**). 5-FU liposomes (FE) not exposed to ultrasound served as control.

**Figure 9 pharmaceutics-13-00821-f009:**
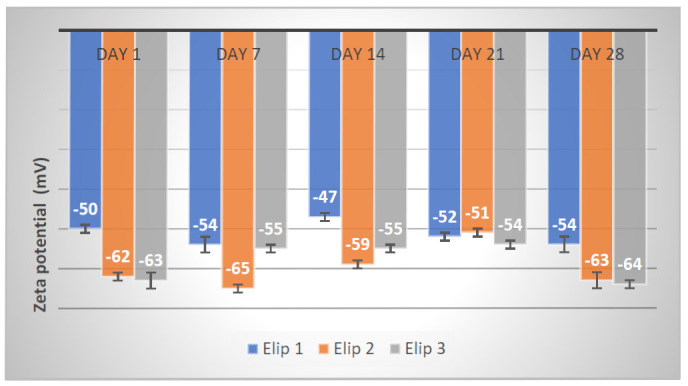
DLS result showing the stability study of echogenic liposomes by zeta potential.

**Table 1 pharmaceutics-13-00821-t001:** Zeta potential (ZP) and encapsulation efficiency (%EE) of the formulations.

Formulations	ZP ± SD (mV)	EE ± SD (%)
*FE 1	−54 ± 1	52 ± 1
FE 2	−56 ± 1	62 ± 2
FE 3	−58 ± 1	51 ± 1
*Elip 1	−62 ± 1	58 ± 1
Elip 2	−63 ± 2	51 ± 7
Elip 3	−59 ± 3	44 ± 8

(*FE 1: Liposomal 5-FU, *Elip 1: Echogenic liposome 5-FU).

## Data Availability

Data are contained within the article or Appendix A. The data presented in this study are available in [Appendix A].

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
