# Peer review of "Ultrasound-Triggered Release of 5-Fluorouracil from Soy Lecithin Echogenic Liposomes"

_pharmaceutics, 2021, doi:10.3390/pharmaceutics13060821_

Round 1
Reviewer 1 Report
The paper of Charles et al. describes the use of 5-fluorouracil (5-FU) entrapped in so-called echogenic
liposomes for the treatment of colorectal cancer. The liposomes are prepared from crude soy lecithin
and cholesterol, and the sensitivity of these liposomes regarding ultrasound is brought about by the
co-entrapment of argon and the application of low-frequency ultrasound. By varying the amplitude
of the ultrasound, the release of the drug could be controlled within certain limits. The results
obtained could contribute to a better, and triggered application of 5-FU in the therapy of colorectal
cancer in the future.
However, after intensive study of the manuscript, I have concluded that this manuscript cannot be
accepted as is due to lack of scientific details in the descriptions and the overall concept of the study.
Let me illustrate this decision with a few examples.
In the introduction, the use of 5-FU is described in detail. However, reference is often made to rather
older literature. The reader might wonder here whether there is not more recent literature in this
field. The authors are encouraged to compare their work to newer references. For example:
- colorectal cancer statistics 2014 – is there anything new?
- ref [7] should be a reference with respect to the drug delivery applications of liposomes in
general – however, ref. [7] is about “Attenuation and Size Distribution Measurements …”.
There are a lot of reviews about this topic!
- Regarding “theranostic”, there are much more papers, for example, Kooiman K, et al. 2020
Ultrasound-Responsive Cavitation Nuclei for Therapy and Drug Delivery, Ultrasound Med.
Biol. 46, 1296-1325.
The Materials and Methods section lack essential details:
- The appropriate equipment for the various examinations should be noted with the respective
methods. This makes it easier for the reader.
- Is the lecithin used in this study even approved for parenteral (topical?) use in humans?
What about the composition of the "crude lecithin" – roughly 7 g are missing in the
composition. Is the composition comparable over several batches? I am of the opinion that
the composition has a significant influence on the release kinetics of the liposomal
formulations - and if a comparable composition of the crude lecithin cannot be guaranteed,
use is not possible.
- 2.2.1. Why is the encapsulation efficiency of 5-FU determined indirectly by gravimetry? Later,
the authors are using a “validated HPLC method to determine the total available
concentration of 5-FU.” (see 2.2.8. and 2.2.9.) I wonder why this method is not used here?
- 2.2.3. Details regarding the DLS measurement are missing. How many runs, what duration,
how many replications of the measurement of the same sample?
- 2.2.4. What EM was used? What technique was used, cryo fixation or negative stain? In the
latter case, what staining agent was used?
- 2.2.5. “at a flow rate of 10°C/min” should be “heating rate”; any more details regarding data
treatment?
- 2.2.6. and 2.2.7. The authors are encouraged to provide more technical details regarding the
measurements: what controls were used? How was the raw data handled until the final
visualization?
- 2.2.8. How can the release from a solution (control) be determined?
Also, in the results and discussion part of the manuscript there are, in my opinion, partly serious
deficiencies in the presentation of the results.
- Table 1. What are the differences between the samples FE1, FE2, and FE3; and further Elip1
and Elip2? Are these merely duplicates of the same composition? If this is the case, why were
only two Elip-duplicates used? Or is the composition different as the phrase "with the mass
ratio of 2:1" (page 6, line 269) suggests? If these are different compositions, then it is
essential to explain them. If not - why were other compositions not investigated?
- 3.2. The statement “in the range of -50 to -60 mV” (page 6, line 288) - The statement
contradicts the data in the table 1. What samples are shown in Fig 1A, 1B, and 1C (page7)?
- 3.3. This EM image does not show liposomes in my opinion! (Moreover, it is very poorly
illuminated during the exposure; I think, the electrode beam was not in focus). About the
shape and appearance of liposomes in EM, please have a look at, for example, Meister A,
Blume A, 2017 (Cryo)Transmission Electron Microscopy of Phospholipid Model Membranes
Interacting with Amphiphilic and Polyphilic Molecules, Polymers 9, 521.
- 3.4. The DSC curves of “empty liposomes” and “Liposomal 5-FU” (Fig.2) are not explained in
the text at all. What is the origin of the endothermic transitions between 100 and 170°C in
both scans?
- 3.5. Here, and the author also notes this, the crystallinity of the active ingredient found in the
XRD somehow contradicts the DSC results. The author attempts to resolve this by stating that
5-FU has a "more amorphous state" (page 8, line 317). But what does this mean? In addition,
I think the figure 3 can be better displayed: limit the display of theta to 10-60°; zoom for
“empty liposomes” and “liposomal 5-FU”.
- 3.6. When describing and assigning the individual vibrations, please be more explicit: are
they valence or deformation vibrations? What “amino group of the phospholipids” (page 9,
lines 330, 331) is meant? Is the reference [34] correct? There are other references regarding
the IR of phospholipids: Snyder RG, Liang GL, Strauss HL, Mendelsohn R, 1996 IR
Spectroscopic Study of the Structure and Phase Behavior of Long-Chain
Diacylphosphatidylcholines in the Gel State, Biophys. J. 71, 3186-3198; or Mendelsohn R,
Moore DJ, 1998 Vibrational spectroscopic studies of lipid domains in biomembranes and
model systems, Chem. Phys. Lipids 96, 141-157 – to name but a few.
What is the origin of the vibrations between 2000 and 2300 cm-1 in “empty liposomes” and
“liposomal 5-FU”?
- 3.7. How can the release from a solution (control) be determined? Is the drug really dissolved
completely? In Figure 5, the error bars are missing. What composition was tested? Is
reference [35] correct in this context?
- 3.8. Reference [36] uses capsules and not liposomes. Are the differences shown in Fig. 6
significant? What about the control experiments in this ultrasound triggered experiments?
Readers will wonder at this point how the other liposomes behave? Does the ultrasound also
influence these liposomes (FE1-3)? I see this comparison as essential (!), otherwise the use of
echogenic liposomes would be redundant.
- Again 3.8. Figures 7A and B. Here, a control is given – however, which control was used in
these experiments, free 5-FU or liposomal 5-FU (FE1-3)? In addition, Figure 7A and 7B could
be combined (displayed next to each other). Again, the error bars are missing in these
figures.
- 3.10. Now why have three (Elip 1-3) been used (compare to table 1)? In addition, the
presentation of the results can be optimized; comparable to the other figures in the
manuscript.
Author Response
"Please see the attachment."

Reviewer 2 Report
The authors present an interesting paper entitled "Ultrasound-Triggered Release of 5-fluorouracil from Soy Lecithin Echogenic Liposomes"
The idea of the authors is to develop 5-FU loaded liposomes with argon gas entrapped in the nanoparticles for sonosensitivity activity. The objective is to improve the therapeutic efficacy of the drug in the treatment of colorectal cancer reducing its resistance and its associated gastrointestinal and bone toxicity. There is also associated a low bioavailability after rectal and oral administration because of the low lipophilicity of the API. Some morphological and physicochemical properties have been evaluated, as well as the biopharmaceutical profiles of these 5-FU nanoparticles. Authors indicate that the results showed in this paper are encouraging for the stimulated and controlled 5-FU release for the treatment of colorectal cancer.
In my opinion, this work is very well structured; the different sections, introduction, material and methods, and results are very well developed and very well described. Although the development of 5-FU nanosystems is not of great originality, this work does provide a novelty associated with the effect of the ultrasound as physical stimuli to control the release of this active.
There are some points that should be discussed:
- The freeze-drying process should be properly described in the section of Materials and methods. This is important to understand the presented results.
- Related particle size analysis, results indicate a wide interval in almost the great mass of the nanoparticles. Could be expected some problems associated with the pharmaceutical profiles of the system, and therefore some deviation from the pre-predicted therapeutic responses? This should be indicated.
- Zeta Potential. Authors reports that formulations showed negative Zeta Potential in the range of −50 to −60 mV, and so, “this electrokinetic property exert a control on the stability of the liposomal formulations under physiological conditions”. Authors also report that no changes have been found in the freeze-dried powdered formulation, being stable when the freeze-dried echogenic liposomal formulations were stored at 4°C for four weeks. These are good results, but in my opinion, experimental data must be presented before assuming that “this electrokinetic property exert a control on the stability of the liposomal formulations under physiological conditions”.
- The several results presented from the DSC, XRD and Fourier-transform infrared spectroscopy studies indicate that 5-FU was successfully encapsulated in the carrier, in its amorphous state. Could this imply some deviation in the stability, physicochemical properties, biopharmaceutical profile or therapeutical response? No information related to this polymorphism aspect of the active have been referenced.
- Results related to the effects of ultrasound amplitude and exposure time on 5-FU liposomes release profiles are interesting, although as indicated by the authors, the overall mechanism and performance of ultrasound-mediated disruption of liposomes bilayers remains to be clarified.
Nevertheless, and precisely because of the results presented, the research associated to this paper needs to be complemented with some in vitro/in vivo experiments, to have more novelty of the results and to increase the interest of this paper. At least, some research related to hemocompatibility, compatibility with cell lines according to this research, some indicator of internalization capacity and activity in the selected cell line ...
I strongly encourage the researchers of this work to continue their research in this regard, in order to enhance the results obtained.
Author Response
"Please see the attachment."

Reviewer 3 Report
Ultrasound-Triggered Release of 5-fluorouracil from Soy Lecithin Echogenic Liposomes
Thank you for the opportunity to referee this Pharmaceutics full-paper, Ultrasound-Triggered Release of 5-fluorouracil from Soy Lecithin Echogenic Liposomes from Rhodes University, Grahamstown, South Africa. I have made but only two suggestions, hopefully not trivial. My bottom-line will be accept this nice full-paper after minor revisions. In order to investigate the therapeutic index, experiments were designed concerning thin-film hydration liposomes containing entrapped argon gas for ultrasound (US, 20 kHz) release. Good work, well written (pity about the References!)
Fine Introduction, good Materials and Methods and Analysis. Ln 260 (2.2.11. Statistical analysis) please state clearly how many technical replicates in each experiment, I assume n = 3 are the scientific replicates. It will help readers if you clarify for each independent experiment. Sadly, something weird has occurred in your R&D, you need to re-check EVERY Reference and its callout, I have looked at 4; ALL are WRONG! Ln 384 Abed is not [37] possibly [27]; [36] who is SMB? cf Ref [27]; ln 449, ref [41], end at [40]!! Whilst I would not dictate, you should consider adding 2 or 3 References to the research output of Prof E Stride, one of the UK (world?) leaders in (micro)Bubbles/ultrasound technology for (anti-cancer) drug delivery systems and diagnostics.
References – see Pharmaceutics House style. Many Journal names require capital letters and (surely?) Journal abbreviations, also too many capitals in the wrong places. Check out and revise carefully ALL the references and their callouts into the House style.
Bottom-line: accept this nice full-paper after minor revisions. Good work, well written (pity about the Reference checking!)
Author Response
"Please see the attachment"

Reviewer 4 Report
The Authors proposed for the Journal Processes the following paper entitled: “ultrasound triggered release of 5-fluoracil from soy lecithin echogenic liposomes”.
I found this paper interesting; indeed, it is characterized by a high scientific soundness.
Concerning the English, there are some spelling mistakes; for this reason, I invite the authors to revise once more their paper in terms of use of English.
This paper deserves to be published after minor/major revisions.
Therefore, I request to address some issues; the list is reported in the following text.
Abstract
Line 15. How does this drug act against cancer?
Line 18. Here it is said that liposomes were investigated. I would say that the effect of… on liposomes was investigated.
Line 20. Explain sonosensitivity.
I also suggest adding an abbreviation list among the Abstract and the Introduction section, to include acronyms such as 5FU, CSL, CRC and more others found into the overall manuscript.
Introduction
Line 31. “any cancer”. Better “a type of cancer”?
Line 35. Remove the useless spaces in this line.
Line 40. “Serious side effects”. Add references.
Lines 55-57. More references about triggered release from liposomes, that have many applications for anti-cancer scopes. There are a lot of works about triggered release from liposomes: this is the future of the production and use of lipidic vesicles, linked to the possibility to induce release after external or internal stimuli. For example, I could suggest using these references, or maybe the authors could choose other ones:
Amstad, E., Kohlbrecher, J., Müller, E., Schweizer, T., Textor, M., & Reimhult, E. (2011). Triggered release from liposomes through magnetic actuation of iron oxide nanoparticle containing membranes. Nano letters, 11(4), 1664-1670.
Oerlemans, C., Deckers, R., Storm, G., Hennink, W. E., & Nijsen, J. F. W. (2013). Evidence for a new mechanism behind HIFU-triggered release from liposomes. Journal of controlled release, 168(3), 327-333.
Trucillo, P., Campardelli, R., & Reverchon, E. (2020). Liposomes: From Bangham to supercritical fluids. Processes, 8(9), 1022.
Schroeder, A., Honen, R., Turjeman, K., Gabizon, A., Kost, J., & Barenholz, Y. (2009). Ultrasound triggered release of cisplatin from liposomes in murine tumors. Journal of controlled release, 137(1), 63-68.
Line 59. Some lines about the variety of liposomes production methods could be added.
Line 62. Are the authors talking about the possibility to insert NO inside liposomes, according to what is reported in the literature? Is this topic treated in Ref 10 ?
Line 77-78. A very nice described application!
Lines 83-85. Do you have current prices data to compare them?
Line 96. Should this be inserted in the Methods section, with reference to the thin layer hydration method?
Line 112. “purchased by” is missing.
Line 150. “1020 g”. why 1020 and not 1000 ? why not rpm? Describe if there was some conversions.
Line 164. Equation should be inserted as a proper equation, not as an image, since the quality of the figure appears low. Please, substitute it.
Lines 174-179. Liposomes were not affected by this procedure? Do you have any references more?
Line 266. I would unify the paragraphs that talk about PSD, zeta potential and encapsulation efficiencies under one single paragraph, using only one table. Indeed, Table 1 could by characterized by number of formulation, mean size plus/minus standard deviations, zeta potential plus/minus standard deviation and encapsulation efficiency plus/minus SD.
Figure 1a is not clear. The y axis could be “by frequency or by intensity”. Numerosity is sometimes missing some data. What does these 3 PSDs curves represent? Maybe 3 runs on the same sample? Concerning which formulation of Table 1? Please, explain to me what was the intention with this figure 1a.
Moreover, Figure 1c reports the PSD by frequency? Meaning intensity? So, uniform this y axis with the y axis of Figure 1a.
Figure 1b. the shape of the liposomes in this figure seems not to be really spherical. Could you give a comment on this?
Figure 5. Is this figure without the application of ultrasound? If yes, specify it also in the caption of this figure.
After Figure 7a and Figure 7b, could it be possible to add Figure 7c with another diagram, comparing drug release after applying different amplitudes for 15 minutes? Then, it could be interesting to compare only 20 % amplitudes curves at 5, 10 and 15 minutes as varying parameter?
Line 343 and Line 344. Maybe “over” could be substituted with “after”.
Figure 8. Description should be part of the caption, not of the figure.
Conclusions.
Line 454. There is a double space in this line.
May the use of argon cause mechanical damages to human body during drug release? If not, provide references.
Thank you.
Author Response
"Please see the attachment."

Round 2
Reviewer 1 Report
The authors have tried to answer all my questions and the manuscript has certainly gained some quality. However, I still think that the quality of the data and its presentation is not sufficient for publication in Pharmaceutics.
There are still major deficiencies (to name but a few):
- Figure 1 – TEM: I am still of the opinion that there are no liposomes to be seen here; these are artefacts! The scattering of light elements (such as H, C and O) is not sufficient to obtain an electron microscopic image without contrast. The publication cited by the authors (Nkanga, C.I., Krause, R.W.M. Encapsulation of Isoniazid-conjugated Phthalocyanine-In-Cyclodextrin-In-Liposomes Using Heating Method. Sci Rep 9, 11485, 2019) uses liposomes with isoniazid grafted zinc (II) phthalocyanine – and here the Zn(II) acts as “heavy” element.
- Figure 4 – FTIR: The authors state that “The peaks observed between 2000 – 2300 cm-1 in the empty liposome and the liposomal 5-FU corresponds to the (N=C=O) isocyanate group stretching vibration.” However, there is no isocyanate group in classical liposomes made from lecithin! So, what is the origin of these bands? I personally think that these bands originate from the ATR crystal …
- FTIR – data handling: How did you remove the water, CO2 bands from the spectra?
- Figures 5 and 7 – error bars. I personally believe that this is not the error resulting from measuring different samples - but a "mathematical" error resulting from measuring the same sample repeatedly!
- What are the concentrations of the liposomes (and the components) in all different methods?
Author Response
Thank you for your valuable comments, please see attached response

Reviewer 4 Report
I saw that the authors responded point by point to all my issues.
I believe that the manuscript has improved much and now deserves to be published in the present version.
I only suggest to extract the raw data from the Figure 1A Particle Size Distribution diagram and create it again using a drawing program. It would gain in quality of this figure.
Author Response
Thank you for your suggestion, we have re-done the analysis. See attached response.

Round 3
Reviewer 1 Report
The authors have again tried to answer my questions and issues raised in my second reply. They also included TEM images of uranyl acetate samples, which I very much welcome. However, I must state again and in all clarity that the interpretation of the data, especially FTIR and TEM, do not convince me in any way and therefore I cannot support a publication in Pharmaceutics of the manuscript in its present form. Let me explain this in more detail.
TEM
I still believe that Figure 1B cannot be shown. There are no liposomes to be seen; the black “aggregates” are artefacts and – as I wrote in my second reply – the scattering of light elements (H, C, O) is not sufficient to obtain an EM image without staining, for examples with uranyl acetate. Therefore, this image must be deleted, as well as the size determination with the software ImageJ (Figure 1C), as this determination is based on counting artefacts.
Regarding both EM images using uranyl acetate stained samples (Figure 2). First, the description of this method is missing in the Material and Methods part of the manuscript. Secondly, with the best will in the world I cannot recognize spherical liposomes in either of the two figures! The white areas are holes in the film; the deep black aggregates are precipitated uranyl acetate. In my opinion, the magnification is far too low to be able to adequately assess the size of liposomes (according to DLS around 100nm, which is 1/20 of the whole scale bar!). I can only identify some collapsed vesicles in Figure 2A and one large lamellar-like structure in Figure 2B (circular structure on the top of the image, >2µm). Furthermore, I cannot agree with the authors' statement “… showed the spherical shape of the 5-FU encapsulated liposomes with little or no fusion or aggregation observed " since this cannot be concluded from the EM pictures shown here.
I would therefore strongly recommend removing all the EM images.
FTIR
The authors showed new FTIR spectra. In the respond, the authors state that IR spectra were recorded from samples in freeze-dried form. First, I wonder at this point how these IR recordings can be interpreted at all. Are the IR data obtained from freeze-dried samples at all comparable with the other data (X-ray, release, etc.)? Why were the IR measurements not carried out with aqueous samples? Second, the exact procedure is again not described in the Material and Methods part of the manuscript. I had already criticised this in my first report!
In my opinion, there are no significant differences in the IR measurement of the empty liposomes and the 5-FU liposomes (Figure 5) that would allow the conclusion of an inclusion of 5-FU in the liposomes. Furthermore, although the assignment of the individual bands to specific molecular vibrations is important and useful, no conclusions are drawn from it. I cannot support the statement that "The characteristic peak at 1660 cm−1 in the free 5-FU corresponding to the C=O stretching vibration shifted to about 1680 cm-1 in the liposomal 5-FU". in my opinion, this shift (apart from the fact whether this is a real shift) is too small. Moreover, a more detailed evaluation would have to be made, since the phospholipids also show a C=O vibration, which would then have to be separated from the C=O vibration originating from 5-FU (using, for example, 13C-labeled lipids).
To put it briefly, I cannot support the statement “The spectral results further show that the drug was successfully encapsulated in the nanoparticles” made by the authors! The IR data shown does not supports this conclusion and should therefore be completely removed from the manuscript.
Lastly, the C−H stretching vibrational bands are at 2850 and 2920 cm−1 for the symmetrical and antisymmetrical vibration, respectively, and not at 2750 cm−1. What is the origin of the broad band between 3000 and 3800 cm−1 (residual traces of water?)?
Figures in general
Finally, a note on figures in general. In my opinion, these are not presented in a uniform manner. Almost every figure has a different size of the labels, sometimes looks compressed or distorted (fig 3 and 8), uses colours (fig 7) and coloured backgrounds (fig 9) that are not necessary, or appears to be a screenshot (fig 1A), etc. I would strongly encourage the authors to recreate all the figures. Of course, I am aware that this does not change the content of the figures - nevertheless, a uniform, clear presentation of the data is recommended.
In conclusion, I cannot recommend the manuscript in its present form for publication and advise a complete revision of the manuscript - possibly including further data, especially in vitro studies, to prove the applicability of these liposomes and make the manuscript interesting for further readers.